# Recycling Privileged Learning and Distribution Matching for Fairness

**Novi Quadrianto**[*]
Predictive Analytics Lab (PAL)
University of Sussex
Brighton, United Kingdom
n.quadrianto@sussex.ac.uk

**Viktoriia Sharmanska**
Department of Computing
Imperial College London
London, United Kingdom
sharmanska.v@gmail.com

## Abstract

Equipping machine learning models with ethical and legal constraints is a serious issue; without this, the future of machine learning is at risk. This paper takes a step forward in this direction and focuses on ensuring machine learning models deliver fair decisions. In legal scholarships, the notion of fairness itself is evolving and multi-faceted. We set an overarching goal to develop a unified machine learning framework that is able to handle any definitions of fairness, their combinations, and also new definitions that might be stipulated in the future. To achieve our goal, we recycle two well-established machine learning techniques, privileged learning and distribution matching, and harmonize them for satisfying multi-faceted fairness definitions. We consider protected characteristics such as race and gender as privileged information that is available at training but not at test time; this accelerates model training and delivers fairness through unawareness. Further, we cast demographic parity, equalized odds, and equality of opportunity as a classical two-sample problem of conditional distributions, which can be solved in a general form by using distance measures in Hilbert Space. We show several existing models are special cases of ours. Finally, we advocate returning the Pareto frontier of multi-objective minimization of error and unfairness in predictions. This will facilitate decision makers to select an operating point and to be accountable for it.

## 1 Introduction

Machine learning technologies have permeated everyday life and it is common nowadays that an automated system makes decisions for/about us, such as who is going to get bank credit. As more decisions in employment, housing, and credit become automated, there is a pressing need for addressing ethical and legal aspects, including fairness, accountability, transparency, privacy, and confidentiality, posed by those machine learning technologies [1, 2]. This paper focuses on enforcing fairness in the decisions made by machine learning models. A decision is fair if [3, 4, 5]: i) it is not based on a protected characteristic [6] such as gender, marital status, or age (**fair treatment**), ii) it does not disproportionately benefit or hurt individuals sharing a certain value of their protected characteristic (**fair impact**), and iii) *given the target outcomes*, it enforces equal discrepancies between decisions and target outcomes across groups of individuals based on their protected characteristic (**fair supervised performance**).

The above three fairness definitions have been studied before, and several machine learning frameworks for addressing each one or a combination of them are available. We first note that one could ensure fair treatment by simply ignoring protected characteristic features, i.e. *fairness through unawareness*. However this poses a risk of unfairness by proxy as there are ways of predicting

---

[*]Also with National Research University Higher School of Economics, Moscow, Russia.

protected characteristic features from other features [7, 8]. Existing models guard against unfairness by proxy by enforcing fair impact or fair supervised performance constraints in addition to the fair treatment constraint. An example of the fair impact constraint is *the 80% rule* (see e.g. [3, 9, 10]) in which positive decisions must be in favour of group B individuals at least 80% as often as in favour of group A individuals for the case of a binary protected characteristic and a binary decision. Another example of the fair impact constraint is a *demographic parity* in which positive decisions of group B individuals must be at the same rate as positive decisions of group A individuals (see e.g. [11] and earlier works [12, 13, 14]).

In contrast to the fair impact that only concerns about decisions of an automated system, the fair supervised performance takes into account, when enforcing fairness, the discrepancy between decisions (predictions) and target outcomes, which is compatible to the standard supervised learning setting. Kleinberg et al. [15] show that fair impact and fair supervised performance are indeed mutually exclusive measures of fairness. Examples of the fair supervised performance constraint are *equality of opportunity* [4] in which the true positive rates (false negative rates) across groups must match, and *equalized odds* [4] in which both the true positive rates and false positive rates must match. Hardt et al. [4] enforce equality of opportunity or equalized odds by post-processing the soft-outputs of an unfair classifier. The post-processing step consists of learning a different threshold for a different group of individuals. The utilization of an unfair classifier as a building block of the model is deliberate as the main goal of supervised machine learning models is to perform prediction tasks for future data as accurately as possible. Suppose the target outcome is correlated with the protected characteristic, Hardt et al.'s model will be able to learn the ideal predictor, which is not unfair as it represents the target outcome [4]. However, Hardt et al.'s model needs to access the value of the protected characteristic for future data. Situations where the protected characteristic is unavailable due to confidentiality or is prohibited to be accessed due to the fair treatment law requirement will make the model futile [5]. Recent work of Zafar et al. [5] propose de-correlation constraints between supervised performance, e.g. true positive rate, and protected characteristics as a way to achieve fair supervised performance. Zafar et al.'s model, however, will *not* be able to learn the ideal predictor when the target outcome is indeed correlated with a protected characteristic.

This paper combines the benefits of Hardt et al.'s model [4] in its ability to learn the ideal predictor and of Zafar et al.'s model [5] in not requiring the availability of protected characteristic for future data at prediction time. To achieve this, we will be building upon recent advances in the use of privileged information for training machine learning models [16, 17, 18, 19]. *Privileged information* refers to features that can be used at training time but will not be available for future data at prediction time. We propose to consider protected characteristics such as race, gender, or marital status as privileged information. The privileged learning framework is remarkably suitable for incorporating fairness, as it learns the ideal predictor and does not require protected characteristics for future data. Therefore, this paper *recycles* the overlooked privileged learning framework, which is designed for accelerating learning and improving prediction performance, for building a fair classification model.

Enforcing fairness using the privileged learning framework alone, however, might increase the risk of unfairness by proxy. Our proposed model guards against this by explicitly adding fair impact and/or fair supervised performance constraints into the privileged learning model. We recycle a distribution matching measure for fairness. This measure can be instantiated for both fair impact (e.g. demographic parity) and fair supervised performance (e.g. equalized odds and equality of opportunity) constraints. Matching a distribution between function outputs (decisions) across different groups will deliver fair impact, and matching a distribution between errors (discrepancies between decisions and target outcomes) across different groups will deliver fair supervised performance. We further show several existing methods are special cases of ours.

## 2 Related Work

There is much work on the topic of fairness in the machine learning context in addition to those that have been embedded in the introduction. One line of research can be described in terms of learning fair models by modifying feature representations of the data (e.g. [20, 10, 21]), class label annotations ([22]), or even the data itself ([23]). Another line of research is to develop classifier regularizers that penalize unfairness (e.g. [13, 14, 24, 11, 5]). Our method falls into this second line of research. It has also been emphasized that fair models could enforce group fairness definitions (covered in the introduction) as well as individual fairness definitions. Dwork et al. and Joseph et al. [25, 26] define

an individual fairness as a non-preferential treatment towards an individual A if this individual is not as qualified as another individual B; this is a continuous analog of fairness through unawareness [23].

**On privileged learning** Vapnik et al. [16] introduce privileged learning in the context of Support Vector Machines (SVM) and use the privileged features to predict values of the slack variables. It was shown that this procedure can provably reduce the amount of data needed for learning an optimal hyperplane [16, 27, 19]. Additional features for training a classifier that will not necessarily be available at prediction time, privileged information, are widespread. As an example, features from 3D cameras and laser scanners are slow to acquire and expensive to store but have the potential to boost the predictive capability of a trained 2D system. Many variants of privileged learning methods and settings have been proposed such as, structured prediction [28], margin transfer [17], and Bayesian privileged learning [18, 29]. Privileged learning has also been shown [30] to be intimately related to Hinton et al.'s knowledge distillation [31] and Bucila et al.'s [32] model compression in which a complex model is learnt and is then replicated by a simpler model.

**On distribution matching** Distribution matching has been explored in the context of domain adaptation (e.g. [33, 34]), transduction learning (e.g. [35]), and recently in privileged learning [36], among others. The empirical Maximum Mean Discrepancy (MMD) [37] is commonly used as the nonparametric metric that captures discrepancy between two distributions. In the domain adaptation setting, Pan et al. [38] use the MMD metric to project data from target and related source domain into a common subspace such that the difference between the distributions of source and target domain data is reduced. A similar idea has been explored in the context of deep neural networks by Zhang et al. [34], where they use the MMD metric to match both the distribution of the features and the distribution of the labels given features in the source and target domains. In the transduction setting, Quadrianto et al. [35] propose to minimize the mismatch between the distribution of function outputs on the training data and on the target test data. Recently, Sharmanska et al. [36] devise a cross-dataset transfer learning method by matching the distribution of classifier errors across datasets.

## 3 The Fairness Model

In this section, we will formalize the setup of a supervised binary classification task subject to fairness constraints. Assume that we are given a set of $N$ training examples, represented by feature vectors $X = \{\mathbf{x}_1, \ldots, \mathbf{x}_N\} \subset \mathcal{X} = \mathbb{R}^d$, their label annotation, $Y = \{y_1, \ldots, y_N\} \in \mathcal{Y} = \{+1, -1\}$, and protected characteristic information also in the form of feature vectors, $Z = \{\mathbf{z}_1, \ldots, \mathbf{z}_N\} \subset \mathcal{Z}$, where $\mathbf{z}_n$ encodes the protected characteristics of sample $\mathbf{x}_n$. The task of interest is to infer a predictor $f$ for the label $y_{\text{new}}$ of an un-seen instance $\mathbf{x}_{\text{new}}$, given $Y$, $X$ and $Z$. However, $f$ cannot use the protected characteristic $Z$ at decision (prediction) time, as it will constitute an *unfair treatment*. The availability of protected characteristic at training time can be used to enforce *fair impact* and/or *fair supervised performance* constraints. We first describe how to deliver fair treatment via privileged learning. We then detail distribution matching viewpoint of fair impact and fair supervised performance. Frameworks of privileged learning and distribution matching are suitable for protected characteristics with binary/multi-class/continuous values. In this paper, however, we focus on a single protected characteristic admitting binary values as in existing work (e.g. [20, 4, 5]).

### 3.1 Fairness through Unawareness: Privileged Learning

In the privileged learning setting [16], we are given training triplets $(\mathbf{x}_1, \mathbf{x}_1^\star, y_1), \ldots, (\mathbf{x}_N, \mathbf{x}_N^\star, y_N)$ where $(\mathbf{x}_n, y_n) \subset \mathcal{X} \times \mathcal{Y}$ is the standard training input-output pair and $\mathbf{x}_n^\star \in \mathcal{X}^\star$ is additional information about a training instance $\mathbf{x}_n$. This additional (privileged) information is only available during training. In our earlier illustrative example in the related work, $\mathbf{x}_n$ is for example a colour feature from a 2D image while $\mathbf{x}_n^\star$ is a feature from 3D cameras and laser scanners. There is no direct limitation on the form of privileged information, i.e. it could be yet another feature representation like shape features from the 2D image, or a completely different modality like 3D cameras in addition to the 2D image, that is specific for each training instance. The goal of privileged learning is to use $\mathbf{x}_n^\star$ to accelerate the learning process of inferring an optimal (ideal) predictor in the data space $\mathcal{X}$, i.e. $f : \mathcal{X} \to \mathcal{Y}$. The difference between accelerated and non-accelerated methods is in the rate of convergence to the optimal predictor, e.g. $1/N$ cf. $1/\sqrt{N}$ for margin-based classifiers [16, 19].

From the description above, it is apparent that both privileged learning model and fairness model aim to use data, privileged feature $\mathbf{x}_n^\star$ and protected characteristic $\mathbf{z}_n$ respectively, that are available at

training time only. We propose to recycle privileged learning model for achieving fairness through unawareness by taking protected characteristics as privileged information. For a single binary protected characteristic $z_n$, $\mathbf{x}_n^\star$ is formed by concatenating $\mathbf{x}_n$ and $z_n$. This is because privileged information has to be instance specific and richer than $\mathbf{x}_n$ alone, and this is not the case when only a single binary protected characteristic is used. By using privileged learning framework, the predictor $f$ is *unaware* of protected characteristic $z_n$ as this information is *not used* as an input to the predictor. Instead, $z_n$, together with $\mathbf{x}_n$, is used to distinguish between easy-to-classify and difficult-to-classify data instances and subsequently to use this knowledge to accelerate the learning process of a predictor $f$ [16, 17]. Easiness and hardness can be defined, for example, based on the distance of data instance to the decision boundary (margin) [16, 17, 19] or based on the steepness of the logistic likelihood function [18]. Our specific choice of easiness and hardness definition is detailed in Section 3.3.

A direct *advantage* of approaching fairness from the privileged lens is the learning acceleration can be used to limit the performance degradation of the fair model as it now has to trade-off two goals: good prediction performance and respecting fairness constraints. An obvious *disadvantage* is an increased risk of unfairness by proxy as knowledge of easy-to-classify and difficult-to-classify data instances is based on protected characteristics. The next section describes a way to alleviate this based on a distribution matching principle.

## 3.2 Demographic Parity, Equalized Odds, Equality of opportunity, and Beyond: Matching Conditional Distributions

We have the following definitions for several fairness criteria [25, 4, 5]:

**Definition A** Demographic parity (fair impact): A binary decision model is fair if its decision $\{+1, -1\}$ are independent of the protected characteristic $z \in \{0, 1\}$. A decision $\hat{f}$ satisfies this definition if

$$P(\text{sign}(\hat{f}(\mathbf{x})) = +1 | z = 0) = P(\text{sign}(\hat{f}(\mathbf{x})) = +1 | z = 1).$$

**Definition B** Equalized odds (fair supervised performance): A binary decision model is fair if its decisions $\{+1, -1\}$ are conditionally independent of the protected characteristic $z \in \{0, 1\}$ given the target outcome $y$. A decision $\hat{f}$ satisfies this definition if

$$P(\text{sign}(\hat{f}(\mathbf{x})) = +1 | z = 0, y) = P(\text{sign}(\hat{f}(\mathbf{x})) = +1 | z = 1, y), \text{ for } y \in \{+1, -1\}.$$

For the target outcome $y = +1$, the definition above requires that $\hat{f}$ has equal *true positive rates* across two different values of protected characteristic. It requires $\hat{f}$ to have equal *false positive rates* for the target outcome $y = -1$.

**Definition C** Equality of opportunity (fair supervised performance): A binary decision model is fair if its decisions $\{+1, -1\}$ are conditionally independent of the protected characteristic $z \in \{0, 1\}$ given the *positive* target outcome $y$. A decision $\hat{f}$ satisfies this definition if

$$P(\text{sign}(\hat{f}(\mathbf{x})) = +1 | z = 0, y = +1) = P(\text{sign}(\hat{f}(\mathbf{x})) = +1 | z = 1, y = +1).$$

Equality of opportunity only constrains equal *true positive rates* across the two demographics.

All three fairness criteria rely on the definition that data across the two demographics should exhibit similar behaviour, i.e. matching positive predictions, matching true positive rates, and matching false positive rates. A *natural* pathway to inject these into any learning model is to use a distribution matching framework. This matching assumption is well founded if we assume that both data $X_{z=0} = \{\mathbf{x}_1^{z=0}, \ldots, \mathbf{x}_{N_{z=0}}^{z=0}\} \subset \mathcal{X}$ and another data $X_{z=1} = \{\mathbf{x}_1^{z=1}, \ldots, \mathbf{x}_{N_{z=1}}^{z=1}\} \subset \mathcal{X}$ are drawn independently and identically distributed from the same distribution $p(\mathbf{x})$ on a domain $\mathcal{X}$. It therefore follows that for any function (or set of functions) $f$ the distribution of $f(\mathbf{x})$ where $\mathbf{x} \sim p(\mathbf{x})$ should also behave in the same way across the two demographics. We know that this is not automatically true if we get to choose $f$ after seeing $X_{z=0}$ and $X_{z=1}$. In order to allow us to draw on a rich body of literature for comparing distributions, we cast the goal of enforcing distributional similarity across two demographics as a *two-sample problem*.

### 3.2.1 Distribution matching

First, we denote the applications of our predictor $\hat{f} : \mathcal{X} \rightarrow \mathbb{R}$ to data having protected characteristic value zero by $\hat{f}(X_{Z=0}) := \{\hat{f}(\mathbf{x}_1^{z=0}), \ldots, \hat{f}(\mathbf{x}_{N_{z=0}}^{z=0})\}$, likewise by $\hat{f}(X_{Z=1}) :=$

$\{\hat{f}(\mathbf{x}_1^{z=1}), \ldots, \hat{f}(\mathbf{x}_{N_{z=1}}^{z=1})\}$ for value one. For enforcing the *demographic parity* criterion, we can enforce the closeness between the distributions of $\hat{f}(\mathbf{x})$. We can achieve this by minimizing:

$$D(\hat{f}(X_{Z=0}), \hat{f}(X_{Z=1})), \text{ the distance between the two distributions } \hat{f}(X_{Z=0}) \text{ and } \hat{f}(X_{Z=1}). \quad (1)$$

For enforcing the *equalized odds* criterion, we need to minimize both

$$D(\mathbb{I}[Y = +1]\hat{f}(X_{Z=0}), \mathbb{I}[Y = +1]\hat{f}(X_{Z=1})) \text{ and } D(\mathbb{I}[Y = -1]\hat{f}(X_{Z=0}), \mathbb{I}[Y = -1]\hat{f}(X_{Z=1})). \quad (2)$$

We make use of Iverson's bracket notation: $\mathbb{I}[P] = 1$ when condition $P$ is true and $0$ otherwise. The first will match true positive rates (and also false negative rates) across the two demographics and the latter will match false positive rates (and also true negative rates). For enforcing *equality of opportunity*, we just need to minimize

$$D(\mathbb{I}[Y = +1]\hat{f}(X_{Z=0}), \mathbb{I}[Y = +1]\hat{f}(X_{Z=1})). \quad (3)$$

To go *beyond* true positive rates and false positive rates, Zafar et al. [5] raise the potential of removing unfairness by enforcing equal *misclassification rates*, *false discovery rates*, and *false omission rates* across two demographics. False discovery and false omission rates, however, with their fairness model are difficult to encode. In the distribution matching sense, those can be easily enforced by minimizing

$$D(1 - Y\hat{f}(X_{Z=0}), 1 - Y\hat{f}(X_{Z=1})), \quad (4)$$

$$D(\mathbb{I}[Y = +1]\max(0, -\hat{f}(X_{Z=0})), \mathbb{I}[Y = +1]\max(0, -\hat{f}(X_{Z=1}))), \text{ and} \quad (5)$$

$$D(\mathbb{I}[Y = -1]\max(0, \hat{f}(X_{Z=0})), \mathbb{I}[Y = -1]\max(0, \hat{f}(X_{Z=1}))) \quad (6)$$

for misclassification, false omission, and false discovery rates, respectively.

**Maximum mean discrepancy** To avoid a parametric assumption on the distance estimate between distributions, we use the Maximum Mean Discrepancy (MMD) criterion [37], a non-parametric distance estimate. Denote by $\mathcal{H}$ a Reproducing Kernel Hilbert Space with kernel $k$ defined on $\mathcal{X}$. In this case one can show [37] that whenever $k$ is characteristic (or universal), the map

$$\mu : p \to \mu[p] := \mathbb{E}_{\mathbf{x} \sim p(\mathbf{x})}[k(\hat{f}(\mathbf{x}), \cdot)] \text{ with associated distance } \text{MMD}^2(p, p') := \|\mu[p] - \mu[p']\|^2$$

characterizes a distribution uniquely. Examples of characteristic kernels [39] are Gaussian RBF, Laplacian and $B_{2n+1}$-splines. With a this choice of kernel functions, the MMD criterion matches infinitely many moments in the Reproducing Kernel Hilbert Space (RKHS). We use an unbiased linear-time estimate of MMD as follows [37, Lemma 14]: $\widehat{\text{MMD}^2} = \frac{1}{N}\sum_i^N k(\hat{f}(\mathbf{x}_{z=0}^{2i-1}), \hat{f}(\mathbf{x}_{z=0}^{2i})) - k(\hat{f}(\mathbf{x}_{z=0}^{2i-1}), \hat{f}(\mathbf{x}_{z=1}^{2i})) - k(\hat{f}(\mathbf{x}_{z=0}^{2i}), \hat{f}(\mathbf{x}_{z=1}^{2i-1})) + k(\hat{f}(\mathbf{x}_{z=1}^{2i-1}), \hat{f}(\mathbf{x}_{z=1}^{2i}))$, with $N := \lfloor\min(N_{z=1}, N_{z=0})\rfloor$.

### 3.2.2 Special cases

Before discussing a specific composition of privileged learning and distribution matching to achieve fairness, we consider a number of special cases of matching constraint to show that many of existing methods use this basic idea.

**Mean matching for demographic parity** Zemel et al. [20] balance the mapping from data to one of $C$ latent prototypes across the two demographics by imposing the following constraint: $\frac{1}{N_{z=0}}\sum_{n=1}^{N_{z=0}} \hat{f}(\mathbf{x}_n^{z=0}; c) = \frac{1}{N_{z=1}}\sum_{n=1}^{N_{z=1}} \hat{f}(\mathbf{x}_n^{z=1}; c); \ \forall c = 1, \ldots, C$, where $\hat{f}(\mathbf{x}_n^{z=0})$ is a softmax function with $C$ prototypes. Assuming a linear kernel $k$ on this constraint is equivalent to requiring that for each c

$$\mu[\hat{f}(\mathbf{x}_n^{z=0}; c)] = \frac{1}{N_{z=0}}\sum_{n=1}^{N_{z=0}} \left\langle \hat{f}(\mathbf{x}_n^{z=0}; c), \cdot \right\rangle = \frac{1}{N_{z=1}}\sum_{n=1}^{N_{z=1}} \left\langle \hat{f}(\mathbf{x}_n^{z=1}; c), \cdot \right\rangle = \mu[\hat{f}(\mathbf{x}_n^{z=1}; c)].$$

**Mean matching for equalized odds and equality of opportunity** To ensure equal false positive rates across the two demographics, Zafar et al. [5] add the following constraint to the training objective

of a linear classifier $\hat{f}(\mathbf{x}) = \langle \mathbf{w}, \mathbf{x} \rangle$: $\sum_{n=1}^{N_{z=0}} \min(0, \mathbb{I}[y_n = -1]\hat{f}(\mathbf{x}_n^{z=0})) = \sum_{n=1}^{N_{z=1}} \min(0, \mathbb{I}[y_n = -1]\hat{f}(\mathbf{x}_n^{z=1}))$. Again, assuming a linear kernel $k$ on this constraint is equivalent to requiring that

$$\mu[\min(0, \mathbb{I}[y_n = -1]\hat{f}(\mathbf{x}_n^{z=0}))] = \frac{1}{N_{z=0}} \sum_{n=1}^{N_{z=0}} \left\langle \min(0, \mathbb{I}[y_n = -1]\hat{f}(\mathbf{x}_n^{z=0})), \cdot \right\rangle$$

$$= \frac{1}{N_{z=1}} \sum_{n=1}^{N_{z=1}} \left\langle \min(0, \mathbb{I}[y_n = -1]\hat{f}(\mathbf{x}_n^{z=1})), \cdot \right\rangle = \mu[\min(0, \mathbb{I}[y_n = -1]\hat{f}(\mathbf{x}_n^{z=1}))].$$

The $\min(\cdot)$ function ensures that we only match false positive rates as without it both false positive and true negative rates will be matched. Relying on *means* for matching both false positive and true negative is not sufficient as the underlying distributions are multi-modal; it motivates the need for *distribution* matching.

### 3.3 Privileged learning with fairness constraints

Here we describe the proposed model that recycles two established frameworks, privileged learning and distribution matching, and subsequently *harmonizes* them for addressing fair treatment, fair impact, fair supervised performance and beyond in a unified fashion. We use SVM$_\Delta$+ [19], an SVM-based classification method for privileged learning, as a building block. SVM$_\Delta$+ modifies the required distance of data instance to the decision boundary based on easiness/hardness of that data instance in the privileged space $\mathcal{X}^\star$, a space that contains protected characteristic $Z$. Easiness/hardness is reflected in the negative of the confidence, $-y_n(\langle \mathbf{w}^\star, \mathbf{x}_n^\star \rangle + b^\star)$ where $\mathbf{w}^\star$ and $b^\star$ are some parameters; the higher this value, the harder this data instance to be classified correctly even in the rich privileged space. Injecting the distribution matching constraint, the final Distribution Matching+ (DM+) optimization problem is now:

$$\underset{\substack{\mathbf{w} \in \mathbb{R}^d, b \in \mathbb{R} \\ \mathbf{w}^\star \in \mathbb{R}^{d^\star}, b^\star \in \mathbb{R}}}{\text{minimize}} \quad \tfrac{1}{2} \underbrace{\|\mathbf{w}\|_{\ell_2}^2}_{\substack{\text{regularisation on model } \textbf{without} \\ \text{protected characteristic}}} + \tfrac{1}{2}\gamma \underbrace{\|\mathbf{w}^\star\|_{\ell_2}^2}_{\substack{\text{regularisation on model } \textbf{with} \\ \text{protected characteristic}}} + C\Delta \underbrace{\sum_{n=1}^{N} \max\left(0, -y_n[\langle \mathbf{w}^\star, \mathbf{x}_n^\star \rangle + b^\star]\right)}_{\text{hinge loss on model } \textbf{with} \text{ protected characteristic}} +$$

$$+ C \underbrace{\sum_{n=1}^{N} \max\left(0, 1 - y_n[\langle \mathbf{w}^\star, \mathbf{x}_n^\star \rangle + b^\star] - y_n[\langle \mathbf{w}, \mathbf{x}_n \rangle + b]\right)}_{\substack{\text{hinge loss on model } \textbf{without} \text{ protected characteristic} \\ \text{but } \textbf{with} \text{ margin dependent on protected characteristic}}} \tag{7a}$$

$$\text{subject to} \quad \overbrace{\text{MMD}^2(p_{z=0}, p_{z=1}) \le \epsilon}^{\text{constraint for removing unfairness by proxy}}, \tag{7b}$$

where $C, \Delta, \gamma$ and an upper-bound $\epsilon$ are hyper-parameters. Terms $p_{z=0}$ and $p_{z=1}$ are distributions over appropriately defined fairness variables across the two demographics, e.g. $\hat{f}(X_{Z=0})$ and $\hat{f}(X_{Z=1})$ with $\hat{f}(\cdot) = \langle \mathbf{w}, \cdot \rangle + b$ for demographic parity and $\mathbb{I}[Y = +1]\hat{f}(X_{Z=0})$ and $\mathbb{I}[Y = +1]\hat{f}(X_{Z=1})$ for equality of opportunity. We have the following observations of the knowledge transfer from the privileged space to the space $\mathcal{X}$ without protected characteristic (refer to the last term in (7a)):

- Very large positive value of the negative of the confidence in the space that includes protected characteristic, $-y_n[\langle \mathbf{w}^\star, \mathbf{x}_n^\star \rangle + b^\star] >> 0$ means $\mathbf{x}_n$, without protected characteristic, is expected to be a *hard-to-classify* instance therefore its margin distance to the decision boundary is *increased*.

- Very large negative value of the negative of the confidence in the space that includes protected characteristic, $-y_n[\langle \mathbf{w}^\star, \mathbf{x}_n^\star \rangle + b^\star] << 0$ means $\mathbf{x}_n$, without protected characteristic, is expected to be an *easy-to-classify* instance therefore its margin distance to the decision boundary is *reduced*.

The formulation in (7) is a *multi-objective* optimization with three competing goals: minimizing empirical error (hinge loss), minimizing model complexity ($\ell_2$ regularisation), and minimizing prediction discrepancy across the two demographics (MMD). Each goal corresponds to a different optimal solution and we have to accept a compromise in the goals. While solving a single-objective optimization implies to search for a single best solution, a collection of solutions at which no goal can be improved without damaging one of the others (*Pareto frontier*) [40] is sought when solving a multi-objective optimization.

**Multi-objective optimization**   We first note that the MMD fairness criteria will introduce non-convexity to our optimization problem. For a non-convex multi-objective optimization, the Pareto frontier may have non-convex portions. However, any Pareto optimal solution of a multi-objective optimization can be obtained by solving the constraint problem for an upper bound $\epsilon$ (as in (7b)) regardless of the non-convexity of the Pareto frontier [40].

Alternatively, the Convex Concave Procedure (CCP) [41], can be used to find an approximate solution of the problem in (7) by solving a succession of convex programs. CCP has been used in several other algorithms enforcing fair impact and fair supervised performance to deal with non-convexity of the objective function (e.g. [24, 5]). However, it was noted in [35] that for an objective function that has an additive structure as in our DM+ model, it is better to use the non-convex objective directly.

## 4   Experiments

We experiment with two datasets: The ProPublica COMPAS dataset and the Adult income dataset. ProPublica COMPAS (Correctional Offender Management Profiling for Alternative Sanctions) has a total of 5,278 data instances, each with 5 features (e.g., count of prior offences, charge for which the person was arrested, race). The binary target outcome is whether or not the defendant recidivated within two years. For this dataset, we follow the setting in [5] and consider race, which is binarized as either black or white, as a protected characteristic. We use $4,222$ instances for training and $1,056$ instances for test. The Adult dataset has a total of $45,222$ data instances, each with 14 features (e.g., gender, educational level, number of work hours per week). The binary target outcome is whether or not income is larger than 50K dollars. For this dataset, we follow [20] and consider gender as a binary protected characteristic. We use $36,178$ instances for training and $9,044$ instances for test.

**Methods**   We have two variants of our distribution matching framework: DM that uses SVM as the base classifier coupled with the constraint in (7b), and DM+ ((7a) and (7b)). We compare our methods with several baselines: support vector machine (SVM), logistic regression (LR), mean matching with logistic regression as the base classifier (Zafar et al.) [5], and a threshold classifier method with protected characteristic-specific thresholds on the output of a logistic regression model (Hardt et al.) [4]. All methods but Hardt et al. do not use protected characteristics at prediction time.

**Optimization procedure**   For our DM and DM+ methods, we identify at least *three* options on how to optimize the multi-objective optimization problem in (7): using Convex Concave Procedure (CCP), using Broyden-Fletcher-Goldfarb-Shanno gradient descent method with limited-memory variation (L-BFGS), and using evolutionary multi-objective optimization (EMO). We discuss those options in turn. *First*, we can express each additive term in the $\widehat{\mathrm{MMD}}^2(p_{z=0}, p_{z=1})$ fairness constraint (7b) as a difference of two convex functions, find the convex upper bound of each term, and place the convexified fairness constraint as part of the objective function. In our initial experiments, solving (7) with CCP tends to ignore the fairness constraint, therefore we do not explore this approach further. As mentioned earlier, the convex upper bounds on each of the additive terms in the MMD constraint become increasingly loose as we move away from the current point of approximation. This leads to the *second* optimization approach. We turn the constrained optimization problem into an unconstrained one by introducing a non-negative weight $C_{\mathrm{MMD}}$ to scale the $\widehat{\mathrm{MMD}}^2(p_{z=0}, p_{z=1})$ term. We then solve this unconstrained problem using L-BFGS. The main challenge with this procedure is the need to trade-off multiple competing goals by tuning several hyper-parameters, which will be discussed in the next section. The CCP and L-BFGS procedures will only return *one* optimal solution from the Pareto frontier. *Third*, to approximate the Pareto-optimal set, we can instead use EMO procedures (e.g. Non-dominated Sorting Genetic Algorithm (NSGA) – II and Strength Pareto Evolutionary Algorithm (SPEA) - II). For the EMO, we also solve the unconstrained problem as in the second approach, but we do *not* need to introduce a trade-off parameter for each term in the objective function. We use the DEAP toolbox [42] for experimenting with EMO.

**Model selection**   For the baseline Zafar et al., as in [5], we set the hyper-parameters $\tau$ and $\mu$ corresponding to the Penalty Convex Concave procedure to $5.0$ and $1.2$, respectively. Gaussian RBF kernel with a kernel width $\sigma^2$ is used for the MMD term. When solving DM (and DM+) optimization problems with L-BFGS, the hyper-parameters $C, C_{\mathrm{MMD}}, \sigma^2,$ (and $\gamma$) are set to $1000., 5000., 10.,$ (and $1.$) for both datasets. For DM+, we select $\Delta$ over the range $\{1., 2., \ldots, 10.\}$

Table 1: Results on multi-objective optimization which balances two main objectives: performance accuracies and fairness criteria. Equal true positive rates are required for ProPublica COMPAS dataset, and equal accuracies between two demographics $z = 0$ and $z = 1$ are required for Adult dataset. The solver of `Zafar et al.` fails on the Adult dataset while enforcing equal accuracies across the two demographics. `Hardt et al.`'s method does not enforce equal accuracies. SVM and LR only optimize performance accuracies. The terms $|\text{Acc.}_{z=0} - \text{Acc.}_{z=1}|$, $|\text{TPR}_{z=0} - \text{TPR}_{z=1}|$, and $|\text{FPR}_{z=0} - \text{FPR}_{z=1}|$ denote accuracy, true positive rate, and false positive rate discrepancies in an absolute term between the two demographics (the smaller the fairer). For ProPublica COMPAS dataset, we **boldface** $|\text{TPR}_{z=0} - \text{TPR}_{z=1}|$ since we enforce the equality of opportunity criterion on this dataset. For Adult dataset, we **boldface** $|\text{Acc.}_{z=0} - \text{Acc.}_{z=1}|$ since this is the fairness criterion.

ProPublica COMPAS dataset (Fairness Constraint on **equal TPRs**)

|  | $|\text{Acc.}_{z=0} - \text{Acc.}_{z=1}|$ | $|\textbf{TPR}_{z=0} - \textbf{TPR}_{z=1}|$ | $|\text{FPR}_{z=0} - \text{FPR}_{z=1}|$ | Acc. |
|---|---|---|---|---|
| LR | 0.0151±0.0116 | **0.2504±0.0417** | 0.1618±0.0471 | 0.6652±0.0139 |
| SVM | 0.0172±0.0102 | **0.2573±0.0158** | 0.1603±0.0490 | 0.6367±0.0212 |
| Zafar et al. | 0.0174±0.0142 | **0.1144±0.0482** | 0.1914±0.0314 | 0.6118±0.0198 |
| Hardt et al.* | 0.0219±0.0191 | **0.0463±0.0185** | 0.0518±0.0413 | 0.6547±0.0128 |
| DM (L-BFGS) | 0.0457±0.0289 | **0.1169±0.0690** | 0.0791±0.0395 | 0.5931±0.0599 |
| DM+ (L-BFGS) | 0.0608±0.0259 | **0.1065±0.0413** | 0.0973±0.0272 | 0.6089±0.0398 |
| DM (EMO Usr1) | 0.0537±0.0121 | **0.1346±0.0360** | 0.1028±0.0481 | 0.6261±0.0133 |
| DM (EMO Usr2) | 0.0535±0.0213 | **0.1248±0.0509** | 0.0906±0.0507 | 0.6148±0.0137 |

*use protected characteristics at prediction time.

Adult dataset (Fairness Constraint on **equal accuracies**)

|  | $|\textbf{Acc.}_{z=0} - \textbf{Acc.}_{z=1}|$ | $|\text{TPR}_{z=0} - \text{TPR}_{z=1}|$ | $|\text{FPR}_{z=0} - \text{FPR}_{z=1}|$ | Acc. |
|---|---|---|---|---|
| SVM | **0.1136±0.0064** | 0.0964±0.0289 | 0.0694±0.0109 | 0.8457±0.0034 |
| DM (L-BFGS) | **0.0640±0.0280** | 0.0804±0.0659 | 0.0346±0.0343 | 0.8152±0.0068 |
| DM+ (L-BFGS) | **0.0459±0.0372** | 0.0759±0.0738 | 0.0368±0.0349 | 0.8127±0.0134 |
| DM (EMO Usr1) | **0.0388±0.0179** | 0.0398±0.0284 | 0.0398±0.0284 | 0.8057±0.0108 |
| DM (EMO Usr2) | **0.0482±0.0143** | 0.0302±0.0212 | 0.0135±0.0056 | 0.8111±0.0122 |

using 5-fold cross validation. The selection process goes as follow: we first sort $\Delta$ values according to how well they satisfy the fairness criterion, we then select a $\Delta$ value at a point before it yields a lower incremental classification accuracy. As stated earlier, we do not need $C, C_{\text{MMD}}, \sigma^2, \gamma, \Delta$ hyper-parameters for balancing multiple terms in the objective function when using EMO for `DM` and `DM+`. There are however several *free* parameters related to the evolutionary algorithm itself. We use the NSGA – II selection strategy with a polynomial mutation operator as in the the original implementation [43], and the mutation probability is set to $0.5$. We do not use any mating operator. We use $500$ individuals in a loop of $50$ iterations (generations).

**Results** Experimental results over $5$ repeats are presented in Table 1. In the ProPublica COMPAS dataset, we enforce equality of opportunity $|\text{TPR}_{z=0} - \text{TPR}_{z=1}|$, i.e. equal true positive rates (Equation (3)), as the fairness criterion (refer to the ProPublica COMPAS dataset in Table 1). Additionally, our distribution matching methods, `DM+` and `DM` also deliver a reduction in discrepancies between false positive rates. We experiment with both L-BFGS and EMO optimization procedures. For EMO, we simulate two decision makers choosing an operating point based on the visualization of Pareto frontier in Figure 1 – Right (shown as `DM (EMO Usr1)` and `DM (EMO Usr2)` in Table 1). For this dataset, `Usr1` has an inclination to be more lenient in being fair for a gain in accuracy in comparison to the `Usr2`. This is actually reflected in the selection of the operating point (see supplementary material). The EMO is run on the $60\%$ of the *training data*, the selection is done on the remaining $40\%$, and the reported results are on the separate test set based on the model trained on the $60\%$ of the training data. The method `Zafar et al.` achieves similar reduction rate in the fairness criterion to our distribution matching methods. As a reference, we also include results of `Hardt et al.`'s method; it achieves the best equality of opportunity measure with only a slight drop in accuracy performance w.r.t. the unfair `LR`. It is important to note that `Hardt et al.`'s method requires protected characteristics at test time. If we allow the usage of protected characteristics at test time, we should expect similar reduction rate in fairness and accuracy measures for other methods [5].

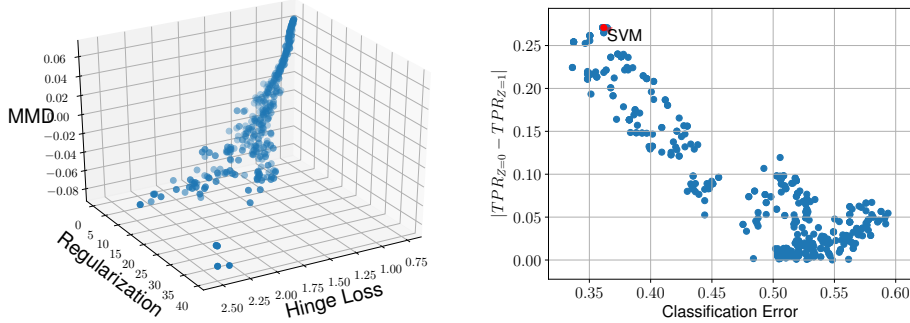

Figure 1: Visualization of a Pareto frontier of our DM method for the ProPublica COMPAS dataset. **Left**: In a 3D criterion space corresponding to the three objective functions: hinge loss, i.e. $\max\left(0, 1 - y_n[\langle \mathbf{w}, \mathbf{x}_n \rangle + b]\right)$, regularization, i.e. $\|\mathbf{w}\|_{\ell_2}^2$, and MMD, i.e. $\widehat{\text{MMD}^2}(p_{z=0}, p_{z=1})$. Fairer models (smaller MMD values) are gained at the expense of model complexity (higher regularization and/or hinge loss values). Note that the unbiased estimate of MMD may be negative [37]. **Right**: The same Pareto frontier but in a 2D space of error and unfairness in predictions. Only the first repeat is visualized; please refer to the supplementary material for the other four repeats, for the Adult dataset, and of the DM+ method.

In the Adult dataset, we enforce equal accuracies |Acc.$_{z=0}$ - Acc.$_{z=1}$| (Equation (4)) as the fairness criterion (refer to the Adult dataset in Table 1). The method whereby a decision maker uses a Pareto frontier visualization for choosing the operating point (DM (EMO Usr1)) reaches the smallest discrepancy between the two demographics. In addition to equal accuracies (Equation (4)), our distribution matching methods, DM+ and DM, also deliver a reduction in discrepancies between true positive and false positive rates w.r.t. SVM (second and third column). In this dataset, Zafar et al. falls into numerical problems when enforcing equal accuracies (vide our earlier discussion on different optimization procedures, especially related to CCP). As observed in prior work [5, 20], the methods that do not enforce fairness (equal accuracies or equal true positive rates), SVM and LR, achieve higher classification accuracy compared to the methods that do enforce fairness: Zafar et al., DM+, and DM. This can be seen in the last column of Table 1.

## 5 Discussion and Conclusion

We have proposed a unified machine learning framework that is able to handle any definitions of fairness, e.g. fairness through unawareness, demographic parity, equalized odds, and equality of opportunity. Our framework is based on learning using privileged information and matching conditional distributions using a two-sample problem. By using distance measures in Hilbert Space to solve the two-sample problem, our framework is general and will be applicable for protected characteristics with binary/multi-class/continuous values. The current work focuses on a single binary protected characteristic. This corresponds to conditional distribution matching with a binary conditioning variable. To generalize this to any type and multiple dependence of protected characteristics, we can use the Hilbert Space embedding of conditional distributions framework of [44, 45].

We note that there are important factors external to machine learning models that are relevant to fairness. However, this paper adopts the established approach of existing work on fair machine learning. In particular, it is taken as given that one typically does not have any control over the data collection process because there is no practical way of enforcing truth/un-biasedness in datasets that are generated by others, such as banks, police forces, and companies.

## Acknowledgments

NQ is supported by the UK EPSRC project EP/P03442X/1 'EthicalML: Injecting Ethical and Legal Constraints into Machine Learning Models' and the Russian Academic Excellence Project '5-100'. VS is supported by the IC Research Fellowship. We thank NVIDIA for GPU donation and Amazon for AWS Cloud Credits. We thank Kristian Kersting and Oliver Thomas for discussions, Muhammad Bilal Zafar for his implementations of [4] and [5], and Sienna Quadrianto for supporting the work.

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
