[Supplementary Material]

# Supplementary Material – Recycling Privileged Learning and Distribution Matching for Fairness

**Novi Quadrianto**[*]
Predictive Analytics Lab (PAL)
University of Sussex
Brighton, United Kingdom
n.quadrianto@sussex.ac.uk

**Viktoriia Sharmanska**
Department of Computing
Imperial College London
London, United Kingdom
sharmanska.v@gmail.com

## 1 Complete experimental results on evolutionary multi-objective

### 1.1 Visualization of a Pareto frontier

Figure 1: ProPublica COMPAS dataset. Visualization of a Pareto frontier of our DM method. In a 3D criterion space corresponding to the three objective functions: hinge loss, i.e. $\max\left(0, 1 - y_n[\langle\mathbf{w}, \mathbf{x}_n\rangle + b]\right)$, regularization, i.e. $\|\mathbf{w}\|_{\ell_2}^2$, and MMD, i.e. $\mathrm{MMD}(p_{Z=0}, p_{Z=1})$.

---

[*]Also with National Research University Higher School of Economics, Moscow, Russia.

Figure 2: ProPublica COMPAS dataset. Visualization of a Pareto frontier of our DM method. The same Pareto frontier as in Figure 1 but in a 2D space of error and unfairness in predictions.

Figure 3: ProPublica COMPAS dataset. Visualization of a Pareto frontier of our DM+ method. For DM+, the criterion space is actually a 5D space, corresponding to the **five** objective functions: two hinge losses, i.e. $\max\left(0, 1 - y_n[\langle \mathbf{w}^\star, \mathbf{x}_n^\star \rangle + b^\star] - y_n[\langle \mathbf{w}, \mathbf{x}_n \rangle + b]\right)$ and $\max\left(0, -y_n[\langle \mathbf{w}^\star, \mathbf{x}_n^\star \rangle + b^\star]\right)$, two regularization terms, i.e. $\|\mathbf{w}\|_{\ell_2}^2$ and $\|\mathbf{w}^\star\|_{\ell_2}^2$, and MMD, i.e. $\mathrm{MMD}(p_{Z=0}, p_{Z=1})$. Only 3D is visualized, corresponding to the MMD, hinge loss $\max\left(0, 1 - y_n[\langle \mathbf{w}^\star, \mathbf{x}_n^\star \rangle + b^\star] - y_n[\langle \mathbf{w}, \mathbf{x}_n \rangle + b]\right)$, and regularisation $\|\mathbf{w}\|_{\ell_2}^2$.

Figure 4: ProPublica COMPAS dataset. Visualization of a Pareto frontier of our DM+ method. The same Pareto frontier as in Figure 3 but in a 2D space of error and unfairness in predictions.

Figure 5: Adult dataset. Visualization of a Pareto frontier of our DM method. In a 3D criterion space corresponding to the three objective functions: hinge loss, i.e. $\max\left(0, 1 - y_n[\langle \mathbf{w}, \mathbf{x}_n \rangle + b]\right)$, regularization, i.e. $\|\mathbf{w}\|_{\ell_2}^2$, and MMD, i.e. $\text{MMD}(p_{Z=0}, p_{Z=1})$.

Figure 6: Adult dataset. Visualization of a Pareto frontier of our DM method. The same Pareto frontier as in Figure 5 but in a 2D space of error and unfairness in predictions.

Figure 7: Adult dataset. Visualization of a Pareto frontier of our DM+ method. For DM+, the criterion space is actually a 5D space, corresponding to the **five** objective functions: two hinge losses, i.e. $\max\left(0, 1 - y_n[\langle \mathbf{w}^\star, \mathbf{x}_n^\star \rangle + b^\star] - y_n[\langle \mathbf{w}, \mathbf{x}_n \rangle + b]\right)$ and $\max\left(0, -y_n[\langle \mathbf{w}^\star, \mathbf{x}_n^\star \rangle + b^\star]\right)$, two regularization terms, i.e. $\|\mathbf{w}\|_{\ell_2}^2$ and $\|\mathbf{w}^\star\|_{\ell_2}^2$, and MMD, i.e. $\mathrm{MMD}(p_{Z=0}, p_{Z=1})$. Only 3D is visualized, corresponding to the MMD, hinge loss $\max\left(0, 1 - y_n[\langle \mathbf{w}^\star, \mathbf{x}_n^\star \rangle + b^\star] - y_n[\langle \mathbf{w}, \mathbf{x}_n \rangle + b]\right)$, and regularisation $\|\mathbf{w}\|_{\ell_2}^2$.

Figure 8: Adult dataset. Visualization of a Pareto frontier of our DM+ method. The same Pareto frontier as in Figure 7 but in a 2D space of error and unfairness in predictions.

## 1.2 Selection of an operating point

Figure 9: ProPublica COMPAS dataset. The above operating point selection is based on $40\%$ of the training data. User 1 has an inclination to be more lenient in being fair for a gain in accuracy in comparison to the User 2.

Figure 10: Adult dataset. The above operating point selection is based on $40\%$ of the training data. In contrast to the selection in ProPublica COMPAS dataset in Figure 9, User 1 now has an inclination to be more strict in being fair and accepts a loss in accuracy in comparison to the User 2.