[Reviews · NeurIPS 2017]

Reviewer 1



This paper proposes a framework which can learn classifiers that satisfy multiple notions of fairness such as fairness through unawareness, demographic parity, equalized odds etc. The proposed framework leverages ideas from two different lines of existing research namely, distribution matching and privileged learning, in order to accommodate multiple notions of fairness. This work builds on two prior papers on fairness - Hardt et. al. and Zafar et. al. in order to create a more generalized framework for learning fair classifiers. The proposed method seems interesting and novel, and the ideas from privileged learning and distribution matching have not been employed in designing fair classifiers so far. The idea of proposing a generalized framework which can handle multiple notions of fairness is quite appealing. The paper, however, has the following weaknesses: 1) the evaluation is weak; the baselines used in the paper are not even designed for fair classification 2) the optimization procedure used to solve the multi-objective optimization problem is not discussed in adequate detail Detailed comments below: Methods and Evaluation: The proposed objective is interesting and utilizes ideas from two well studied lines of research, namely, privileged learning and distribution matching to build classifiers that can incorporate multiple notions of fairness. The authors also demonstrate how some of the existing methods for learning fair classifiers are special cases of their framework. It would have been good to discuss the goal of each of the terms in the objective in more detail in Section 3.3. The part that is probably the most weakest in the entire discussion of the approach is the discussion of the optimization procedure. The authors state that there are different ways to optimize the multi-objective optimization problem they formulate without mentioning clearly which is the procedure they employ and why (in Section 3). There seems to be some discussion about the same in experiments section (first paragraph) and I think what was done is that the objective was first converted into unconstrained optimization problem and then an optimal solution from the pareto set was found using BFGS. This discussion is still quite rudimentary and it would be good to explain the pros and cons of this procedure w.r.t. other possible optimization procedures that could have been employed to optimize the objective. The baselines used to compare the proposed approach and the evaluation in general seems a bit weak to me. Ideally, it would be good to employ baselines that learn fair classifiers based on different notions (E.g., Hardt et. al. and Zafar et. al.) and compare how well the proposed approach performs on each notion of fairness in comparison with the corresponding baseline that is designed to optimize for that notion. Furthermore, I am curious as to why k-fold cross validation was not used in generating the results. Also, was the split between train and test set done randomly? And, why are the proportions of train and test different for different datasets? Clarity of Presentation: The presentation is clear in general and the paper is readable. However, there are certain cases where the writing gets a bit choppy. Comments: 1. Lines 145-147 provide the reason behind x*_n being the concatenation of x_n and z_n. This is not very clear. 2. In Section 3.3, it would be good to discuss the goal of including each of the terms in the objective in the text clearly. 3. In Section 4, more details about the choice of train/test splits need to be provided (see above). While this paper proposes a useful framework that can handle multiple notions of fairness, there is scope for improving it quite a bit in terms of its experimental evaluation and discussion of some of the technical details.

Reviewer 2



The author(s) of this paper take an interesting perspective on the problem of enforcing fairness in learning via the use of the privileged learning framework. Roughly speaking, they allow the protected attributes to be used at training time, but do not allow them to be used at testing time. The way in which this idea prevents proxy variables from affecting fairness is by enforcing a distribution similarity constraint on the outcomes as appropriate for the fairness measure. The authors compare their results to using standard SVMs and show that they can obtain better accuracy with respect to group discrepancies. I think this idea of viewing protected attributes as privileged is interesting. It supports some arguments in the field that the learner should NOT ignore protected attributes when training. And the use of distribution constraints helps block proxy variables, which is important. However, it seems to me that the correct comparison to show that this method is effective is against *other* fairness preserving methods that don't use protected data (or eliminate it) at training time, so as to argue that this approach is better. I suspect that this method might yield better fairness-accuracy tradeoffs as well, but one needs some evidence for this. Comparing to a vanilla SVM seems a little too easy. This is my main hesitation with this paper. The authors are somewhat selective in their citing of prior work. Beyond one stray reference to the Ruggieri et al paper, they ignore a fair amount of work on demographic parity in the pre-2015 era (!), mostly by Calders, Kamishima, and others. The authors also mention disparate impact and the 80% rule, but don't mention the Zafar et al paper in this context (although they cite it for other reasons). They don't cite the Feldman et al KDD 2015 paper which first studied the disparate impact measure in a learning context. Finally, while this is not directly related to the authors' work, they might want to look at the literature on auditing black box models. In particular, the work by Adler et al from ICDM 2016 starts with a classifier that's built in a black box way presumably using protected data, and looks at how to modify the *test* data so as to ensure fairness. This is again an example of protected data being used at training time but not at test time.

Reviewer 3



This paper proposes a unified framework for balancing three studied aspects of fairness in machine learning: fair treatment, where decisions are not based on protected attributes; fair impact, where a certain protected attribute does not end up positively or negatively affecting a data point; and fair supervised performance, where (roughly speaking) every protected group is harmed or helped in the same way. The crux of the paper is putting each of these three competing aspects into a multi-objective optimization problem (Eq 7), either in the objective or as a hard constraint. The authors (I think rightfully for this application area) recommend returning a Pareto frontier of options, instead of setting exogenously parameters to balance the multiple chunks of the objective. The second major contribution of the paper is in determining a better method for dealing with equalized odds/equalized opportunity. Overall, I like the focus of the paper -- the fairness in ML literature is a bit fractured when it comes to who likes which criterion (or criteria), and I'm not convinced even the three aspects from the literature used in this paper are the end-all-be-all of fairness concerns in classification. In terms of core ML / core optimization contributions, the paper is weak. However, Eq 7 seems eminently usable in practice to me, and while I would've liked to see substantially heavier experimental evaluation (because, when it comes down to it, those will make a break the usefulness of any "unification" like this via multi-objective optimization), I think the paper is well-written and has promise.